# Inflammatory and Humoral Immune Responses to Commercial Autogenous *Salmonella* Bacterin Vaccines in Light-Brown Leghorn Pullets: Primary and Secondary Vaccine Responses

**DOI:** 10.3390/vaccines13030311

**Published:** 2025-03-13

**Authors:** Chrysta N. Beck, Jossie M. Santamaria, Gisela F. Erf

**Affiliations:** Department of Poultry Science, Division of Agriculture, University of Arkansas System, Fayetteville, AR 72701, USA; jmsantam@uark.edu

**Keywords:** *Salmonella* vaccine, chicken, inflammation, innate immunity, antibody, T cells

## Abstract

Background/Objectives: Commercial poultry flocks undergo *Salmonella* vaccinations to manage salmonellosis outbreaks. Due to reports of severe injection site reactions to *Salmonella* bacterins, assessment of local inflammatory responses is necessary. The objective was to assess local inflammatory and systemic humoral immune responses to commercial autogenous *Salmonella* bacterin vaccines (SV1 or SV2) following primary or secondary intradermal (i.d.) vaccination in Light-Brown Leghorns (LBLs). Methods: LBL pullets received primary (14 wks) or secondary (19 wks) vaccination by i.d. growing feather (GF) pulp injection of SV1, SV2, *Salmonella* Enteritidis (SE) lipopolysaccharide (LPS), or water–oil–water emulsion (V). Local leukocyte levels and relative cytokine mRNA expression were monitored before (0 d) and at 6 h, 1 d, 2 d, 3 d, 5 d, and 7 d post-GF pulp injection (p.i.). Blood was collected through 28 d post-primary or -secondary vaccination, and SE-specific antibodies were quantified via ELISA. Results: Primary vaccine administration increased local heterophil and macrophage levels and increased IL-6 and IL-8 mRNA expressions at 6 h p.i., independent of treatment. Secondary administration extended these local immune activities through 3 d p.i. and included prolonged IL-17A mRNA expression. Primary and secondary GF-pulp injection with V resulted in rapid lymphocyte recruitment by 6 h p.i., comprised primarily of CD4^+^ and γδ T cells. SV1 and SV2 also produced a T-dependent systemic humoral immune response, as indicated by the IgM-to-IgG isotype switch, along with a memory phenotype in the secondary response. Conclusions: These commercial-killed *Salmonella* vaccines, when prepared in water–oil–water emulsions, stimulated prolonged innate and T helper (Th) 17-type inflammatory responses at the injection site and produced a classic systemic humoral immune response after a second vaccination. Further research is needed to determine if extended inflammation influences adaptive immune responses in eliminating *Salmonella* infection.

## 1. Introduction

For the last three decades, commercial poultry flocks have been vaccinated with autogenous live attenuated and killed bacterin *Salmonella* vaccines to control *Salmonella* load in live birds. Generally, these vaccination programs generally consist of two live attenuated *Salmonella* vaccinations by oral inoculation followed by an intramuscular booster vaccination with a killed *Salmonella* bacterin [1]. Many studies found that these vaccination programs can reduce *Salmonella* colonization in the bird’s gastrointestinal tract and can induce a strong systemic humoral immune response in the weeks following vaccine administration [2,3,4,5,6,7]. For example, Methner [8] determined that vaccination with a live *Salmonella* vaccine at the day of hatch followed by a booster at 28, 35, or 42 days of age produces a strong humoral response with a significant reduction in *Salmonella* load in the ceca and liver following a live *S*. Enteritidis challenge at 56 days of age. Bi- or tri-valent live attenuated *Salmonella* vaccines have been proven to effectively colonize in the gastrointestinal system of birds [9], stimulate the humoral immune system by increasing *Salmonella*-specific antibody production [3,10], reduce colonization of wild-type *Salmonella* serovars in the gut and oviduct [4,11], and decrease fecal shedding of wild-type *Salmonella* [12,13].

By conducting an additional booster vaccination with killed *Salmonella* bacterin, flocks can be protected against specific *Salmonella* serovars. There is evidence that immunological memory produced by killed *Salmonella* vaccines is serovar-specific, and cross-serovar protection after *Salmonella* bacterin vaccination seems limited [14,15]. Even so, Young et al. [14] observed that vaccination of broiler breeder hens with a cocktail of killed autogenous *Salmonella* (serovars Kentucky, Heidelberg, and Hadar) suspended in an oil emulsion adjuvant vaccine significantly reduced the colonization of these serovars when birds were challenged. However, there are cases of severe injection site reactions following intramuscular administration of bacterin vaccines suspended in oil emulsions, with lesions exhibiting high innate leukocyte infiltration, tissue hemorrhaging and caseation, and granuloma development [16,17,18,19,20]. For example, commercial egg-type chickens that were vaccinated intramuscularly in the leg with *Mycoplasma gallisepticum* bacterin suspended in an oil adjuvant resulted in significant swelling and granuloma development in the connective tissue, which was exasperated when birds were transported to a different housing facility [17]. Lesions can also be observed when chickens are intramuscularly vaccinated into the breast muscle tissue with a bacterin vaccine suspended in an oil adjuvant, where lesions are dominated by macrophages with regions of lymphocyte aggregation [19]. Concerningly, case studies have reported that severe injection site reactions to *Salmonella* bacterins in oil adjuvants are also associated with hepatopathy, splenomegaly, pulmonary edema, and renal atrophy in some broiler breeder pullet flocks [21].

Even so, data are limited regarding local cellular responses to *Salmonella* bacterin vaccines formulated in oil emulsions. The minimally invasive growing feather (GF) pulp cutaneous bioassay is a well-established model for testing and monitoring tissue/cellular responses to immunostimulatory materials in avian species [22,23,24,25,26,27,28,29]. Briefly, the pulp of a GF is a skin derivative and complex vascularized tissue capable of sensing and responding to antigens. For longitudinal studies, test material is injected into the dermis of multiple GF pulps. GFs are subsequently collected at various times post-injection for laboratory analyses of pulp tissue (leukocyte identification by immunofluorescent staining by flow cytometry, cytokine mRNA expression by RT-qPCR). This test system not only enables the monitoring of local inflammatory responses to a particular test material, but it also allows for the assessment of systemic humoral immune responses in the form of antigen-specific antibody production measurable in the peripheral blood [27,28,29].

In a recent study assessing local cellular immune responses to a primary intradermal (i.d.) GF-pulp administration of high or low concentrations of autogenous *Salmonella* vaccines or vaccine components, Santamaria et al. [25] observed that the percentage of innate leukocytes (heterophils, macrophages) in GF pulps peaked at 6 h p.i. and decreased to levels above those at pre-injection by 3 d p.i. Primary i.d. GF-pulp injection with an autogenous *Salmonella* bacterin also increased levels of SE-specific IgM and IgY (avian IgG) in the peripheral blood by 5- and 7-days post-i.d. GF-pulp injection, respectively [15]. Interestingly, in that study, observed massive T cell recruitment at 6 h p.i. and B cell recruitment at 24 h p.i. following i.d. injection of a water–oil–water emulsion vehicle alone [25]. Even so, Santamaria et al. [25] did not observe any incidences of severe injection site reactions following a primary administration with high or low doses of autogenous *Salmonella* bacterin vaccine or vaccine components.

Several questions remained upon the conclusion of the work by Santamaria et al. [25], especially when considering the cases of severe injection site reactions observed in the poultry field after intramuscular booster vaccination with *Salmonella* bacterin. These inquiries were as follows: (1) How does previous sensitization with a *Salmonella* bacterin vaccine (similar to what is conducted on commercial poultry farms) influence local cellular activities and the circulating levels of SE-specific antibodies following a secondary i.d. GF-pulp injection? (2) What is the associated leukocyte activity (relative mRNA expression of cytokines) at the site of injection? (3) What are the T cell subpopulations that localized at the site of vaccine vehicle injection?

Therefore, the objective of this study was to investigate temporal, quantitative, and qualitative characteristics of local cellular (GF pulp) and systemic humoral (peripheral blood) immune responses to primary and secondary i.d. GF-pulp injections with autogenous *Salmonella* bacterin vaccines or vaccine components (SE, LPS, or water–oil–water emulsion vehicle) in Light-Brown Leghorn (LBL) pullets.

## 2. Materials and Methods

### 2.1. Experimental Animals

Two groups of 14-weeks-old Light-Brown Leghorn (LBL) female chickens (pullets) maintained by Dr. G. F. Erf were housed at the University of Arkansas System Division of Agriculture (UADA) Poultry Research Farm in Fayetteville, AR, USA were used for this study. Pullets were raised in a floor pen on wood shavings in a biosecure, temperature- and light-controlled environment at the UADA Poultry Research Farm. Water and feed that met or exceeded nutritional requirements were provided ad libitum. The University of Arkansas Institutional Animal Care and Use Committee (IACUC) approved all procedures and protocols involving animals in this experiment (IACUC #21035).

### 2.2. Experimental Design

Two groups of LBL pullets were evaluated for local inflammatory responses to autogenous killed *Salmonella* bacterin vaccines after primary and secondary vaccination (Table 1). Group 1 consisted of 14-week-old LBL pullets that underwent a primary i.d. GF-pulp injection of four autogenous vaccine treatments (trt): SV1 = *Salmonella* Vaccine 1 (180,000 endotoxin units/mL; *n* = 4 birds/trt), SV2 = *Salmonella* Vaccine 2 (68,000 endotoxin units/mL; *n* = 4 birds/trt), LPS = SE lipopolysaccharide (180,000 endotoxin units/mL; *n* = 4 birds/trt), and V = water–oil–water emulsion vaccine vehicle used for SV1, SV2, and LPS vaccine treatments (*n* = 3 bird/trt; 10 μL/GF-pulp, 12 GF/pullet, 120 μL of the total injection volume of the vaccine trt/bird).

Group 2 consisted of 14-week-old LBL pullets that underwent a primary s.c. vaccination (0.5 mL) in the nape of the neck with SV1 (*n* = 4 birds/trt), SV2 (*n* = 4 birds/trt), LPS (*n* = 4 birds/trt), or V (*n* = 3 birds/trt) followed by secondary i.d. GF-pulp injections with the pullet’s respective primary vaccination treatment (10 μL/GF-pulp, 21 GF/bird, 210 μL total injection volume of vaccine trt/bird) five weeks later (19 weeks old).

Vaccines (SV1 and SV2) and vaccine components (LPS and V) contained proprietary formulations (Elanco Animal Health Inc., Indianapolis, IN, USA), and all vaccine trt were ready to use upon receipt. Both vaccines contained several autogenous *Salmonella* serovars, including *S*. Enteritidis.

### 2.3. Sample Collection

Following administration of *Salmonella* vaccine treatments in Groups 1 and 2, 1.5 mL of heparinized blood was collected from the brachial vein prior to vaccination/injection (0 d) and at 3, 5, 7, 10, 14, 21, and 28 d post-primary and -secondary vaccine administrations (*n* = 3–4 pullets/treatment at each timepoint). Heparinized blood was centrifuged at 10,000× *g* for 3 min at room temperature, and plasma samples were stored at −80 °C until the determination of SE-specific antibody levels.

Following primary i.d. GF-pulp injection (Group 1), two GFs were collected before (0 d) and at 6 h, 1 d, 2 d, and 3 d post-i.d. GF-pulp injection (*n* = 4 birds/treatment at each timepoint). Following secondary i.d. GF-pulp injection (Group 2), two GFs were collected before (0 d) and at 6 h, 1 d, 2 d, 3 d, 5 d, and 7 d post-i.d. GF-pulp injection (*n* = 4 birds/treatment at each timepoint). For Groups 1 and 2, one GF pulp was placed in ice-cold Dulbecco’s phosphate-buffered saline (DPBS) and kept on ice until processing for same-day cell population analysis, while the other GF was flash-frozen in liquid nitrogen for RNA isolation and quantitative RT-PCR.

### 2.4. Quantification of S. Enteritidis-Specific Immunoglobulins in Blood Plasma

SE-specific antibody quantification was conducted by ELISA with minor alterations to previously established protocols [25,30,31,32] and detailed fully by Santamaria et al. [25]. Briefly, 96-well flat-bottom plates (#651001, Greiner Bio-One, Kremsmünster, Austria) were coated with 0.2% formalin-inactivated SE at 10^7^ cells/mL. For optimal SE-specific avian IgG (IgY) and IgM detection, blood plasma was diluted at 1:1000 (primary response) and 1:2000 (secondary response), and horse radish peroxidase (HRP)-conjugated goat-anti-chicken (gac)-IgG and -IgM were diluted at 1:20,000. For optimal SE-specific IgA detection, blood plasma was diluted at 1:100 (primary and secondary responses), and HRP-gac-IgA was diluted at 1:10,000. Upon determining optimal dilutions for all reagents and plasma samples, ELISA procedures proceeded (previously detailed [25]). To determine the binding of HRP-conjugated detection antibody, 3,3′,5,5′-tetramethylbenzidine (TMB) substrate was added to each well, and the color change reaction was halted by adding 2 M sulfuric acid. Optical density (OD) was measured by a 96-well plate spectrophotometer (ELx 800, Biotek, Winooski, VT, USA) at 450 nm. Data were adjusted according to plasma sample dilution, and plate-to-plate variation was standardized with the OD of serially diluted pooled plasma samples included in each plate. Controls were present on each plate. Data are presented as absorbance units (a.u.).

### 2.5. Assessing Leukocyte Recruitment and Cytokine mRNA Expression in GF Pulps

#### 2.5.1. Preparation of Cell Suspensions and Cell Population Analysis by Flow Cytometry

Single-cell GF-pulp suspensions and cell population analysis procedures were previously established [24,27,28]. Briefly, GF pulps were extracted from their feather sheath, and each pulp was incubated in 0.5 mL 0.1% collagenase IV-dispase II solution (Collagenase IV, CAS# 9001-12-1, Sigma Aldrich, Saint Louis, MO; Dispase II, Product code 04942078001, Roche Diagnostics Deutschland GmbH, Mannheim, Germany) at 37 °C for 15 min. After incubation, pulps were pushed through a 60-µm nylon mesh with ice-cold DPBS. Suspensions were immunofluorescently stained using a three-color direct staining method with fluorescence-conjugated, chicken leukocyte-specific, mouse-anti-chicken (mac) IgG1 monoclonal antibodies (Table 2 [25]; Southern Biotechnology Associates, Inc., Birmingham, AL). The heterophil (avian neutrophil) population was characterized by size (forward scatter) and granularity (side scatter) of the CD45^+^ cell population [33]. Total lymphocytes were quantified by adding together the percentages of CD4^+^ T cells, CD8α^+^ T cells, CD8α^−^ γδ T cells, and B cells. Negative isotype staining (mouse IgG1 conjugated with FITC, PE, or SPRD) and positive staining controls (mac-CD45 conjugated FITC, PE, or SPRD) were included in each assay for each timepoint. Cell population data were acquired using a BD Accuri C6+ flow cytometer (Becton Dickinson, San Jose, CA, USA), and cell population analysis was conducted using FlowJo software v.10.5 (FlowJo, LLC, Ashland, OR, USA). Data are presented as percentages (%) of total GF-pulp cells.

#### 2.5.2. Total RNA Isolation, cDNA Synthesis, and Relative Gene Expression Analysis

Relative cytokine mRNA expression in i.d.-injected GF pulps was measured according to previously established procedures [25,27,28]. Briefly, injected GF pulps were thawed to 4 °C and aseptically extracted from their feather sheaths. Each GF-pulp was placed in a 1.5 mL safe-lock microcentrifuge tube (Eppendorf, Hauppauge, NY, USA) containing 200 µL TriReagent (Zymo Research Corp., Irvine, CA, USA) and approximately 16, 0.5 mm zirconium oxide beads (Next Advance, Troy, NY, USA). GF-pulp tissues were homogenized twice with the Bullet Blender (Next Advance, Troy, NY, USA) set at Speed 12 for 5 min at 4 °C. Total RNA was isolated from each sample by in-column DNAse digestion using the Directzol kit (Zymo Research Corp., Irvine, CA, USA) according to manufacturer’s procedures. The quality and concentration of extracted total RNA were determined by NanoDrop (Thermo Fisher Scientific, Waltham, MA, USA).

Total RNA was transcribed to cDNA using the High-Capacity cDNA kit with MultiScribe Reverse Transcriptase following manufacturer procedures (Thermo Fisher Scientific, Waltham, MA, USA). Each sample was equalized to 10 ng/µL cDNA using nuclease-free water. Samples were stored at −20 °C until quantitative RT-PCR.

Primer-probe assays for 11 different target genes were used to assess for relative gene expression analysis (Appendix A). All reaction assays were prepared according to Taqman Universal Master Mix with UNG protocols (Thermo Fisher Scientific, Waltham, MA, USA) using final working solution concentrations of 10 ng cDNA, 0.1 µmol/L probe, and 0.1 µmol/L forward and reverse primers per 10 μL reaction. Quantitative RT-PCR was performed in a 7500 Fast Real-Time PCR System (Thermo Fisher Scientific, Waltham, MA, USA), where each target assay-sample reaction was run in duplicate. The temperature profile for each plate run was as follows: (1) 50 °C hold stage for 2 min, (2) 95 °C hold stage for 10 min, and (3) 40 cycles of denaturation (95 °C for 15 s) and annealing (60 °C for 1 min). Cycle threshold (CT) values for each sample target were normalized with the housekeeping gene 28S rRNA (∆CT) [34] and data are shown as 40 − ∆CT [25,35].

### 2.6. Statistical Analysis

Data were analyzed using Sigma Plot 14.5 software (Systat Software Inc., San Jose, CA, USA). Individual pullets were the experimental unit with four pullets per treatment (SV1, SV2, LPS; V = 3 pullets) per data collection timepoint. For leukocyte profile analysis and relative mRNA expression of cytokines in GF-pulp, data were analyzed by two-way ANOVA to determine main effects of the treatment, main effects of time, and treatment by time interactions using Fishers LSD method for multiple means comparison. For SE-specific antibodies in blood plasma, data were analyzed by two-way repeated-measures ANOVA to determine main effect of the treatment, main effects of the time, and treatment by time interactions using Bonferroni *t*-tests for multiple means comparison. Differences were considered significant at *p* ≤ 0.05 for all analyses.

## 3. Results

### 3.1. Salmonella Enteritidis-Specific Antibody Levels in Peripheral Blood Plasma

Following a primary s.c. vaccination and a secondary i.d. GF-pulp injection with *Salmonella* bacterin vaccines or vaccine components (Group 2), SE-specific IgM, IgY (avian IgG), and IgA in blood plasma were quantified by ELISA. Main effect of the treatment s, time main effects, and treatment by time interactions were observed in the three antibody isotypes (*p* < 0.05).

#### 3.1.1. *Salmonella* Enteritidis-Specific IgM

Following primary s.c. vaccination in 14-week-old pullets, there were treatment by time interactions for SE-specific IgM levels in the blood plasma (*p* < 0.001; Figure 1). There was no difference between treatments at pre-vaccination (0 d), 3, 5, and 7 d post-vaccination. SV1- and SV2-vaccinated pullets had greater SE-specific IgM (SE-IgM) levels when compared to LPS- and V-vaccinated pullets at 10 d post-vaccination (p.v. SV1-vaccinated pullets had greater SE-IgM levels than all other treatments at 14, 21, and 28 d p.v. Over time, SV1-vaccinated pullets reached maximal SE-IgM levels at 10, 14, 21, and 28 d p.v., SV2-vaccinated pullets at 10 and 14 d p.v., and SE-IgM levels did not change over time in LPS- or V-vaccinated pullets.

Following a secondary i.d. GF-pulp vaccination in 19-week-old pullets, there were treatment by time interactions for SE-IgM levels in blood plasma (*p* = 0.001; Figure 1). There was no difference in SE-IgM levels between treatments before (0 d) or at 3 d p.v. At 5 d p.v., SV1-vaccinated pullets had greater SE-IgM levels than V-injected pullets but were not different from SV2- or LPS-vaccinated pullets. At 7 d p.v., SV1- and SV2-vaccinated pullets had greater SE-IgM levels than LPS- and V-vaccinated pullets. At 10 d p.v., SV1-vaccinated pullets had greater SE-IgM levels than LPS- or V-vaccinated pullets but were not different from SV2-vaccinated pullets. At 14 d p.v., SE-IgM levels in SV1-vaccinated pullets were greater than all the other treatments, while SV2- and LPS-vaccinated pullets had greater SE-IgM levels than V-injected pullets. There was no difference in SE-IgM levels between treatments at 21 or 28 d p.v. Over time, SV1-injected pullets reached maximal SE-IgM levels by 14 d p.v., SV2-injected pullets by 7 d p.v., and LPS-injected pullets on 5 and 7 d p.v., and there was no change in SE-IgM levels in V-injected pullets.

#### 3.1.2. *Salmonella* Enteritidis-Specific IgY (Avian IgG)

Following primary s.c. vaccination in 14-week-old LBL pullets, there were treatment by time interactions for SE-specific IgG (SE-IgG) levels in blood plasma (*p* < 0.001; Figure 1). There was no difference between treatments pre-vaccination (0 d), and at 3, 5, 7, or 10 d p.v. At 14 and 21 d p.v., SE-IgG levels of SV1-vaccinated pullets were greater compared to LPS- or V-vaccinated pullets but were not different from SV2-vaccinated pullets. At 28 d p.v., SV1-vaccinated pullets had greater SE-IgG levels than all other treatments, and SV2-vaccinated pullets were greater than that in LPS- or V-vaccinated pullets. Over time, SE-IgG plasma levels gradually increased in SV1- and SV2-vaccinated pullets, reaching maximal levels by 28 d p.v., while SE-IgG levels did not change in LPS- or V-vaccinated pullets over time.

Following a secondary i.d. GF-pulp vaccination in 19-week-old LBL pullets, there were time (*p* = 0.049) and treatment (*p* < 0.001) main effects for SE-IgG levels in blood plasma (Figure 1). Regardless of treatment, SE-IgG reached maximal levels at 7, 10, and 14 d p.v., although levels were not different from any of the other timepoints. Overall, SV1- and SV2-injected pullets had greater SE-IgG levels than LPS- or V-injected pullets.

#### 3.1.3. *Salmonella* Enteritidis-Specific IgA

Following primary s.c. vaccine administration, there were treatment by time interactions for SE-specific IgA (SE-IgA) levels in the blood plasma of 14-week-old LBL pullets (*p* = 0.03; Figure 1). SE-IgA levels did not differ between treatments before (0 d) or at 3, 5, 7, and 10 d p.v. At 14 d p.v., SE-IgA levels in SV1-vaccinated pullets were greater than with LPS- or V-vaccination but not different from SV2-vaccinated pullets. At 21 d p.v., pullets from all other treatments had greater SE-IgA levels than V-vaccinated pullets. At 28 d p.v., SE-IgA levels in SV1-vaccinated pullets were greater than in LPS- or V-vaccinated pullets, but not different from SV2-vaccinated pullets. Over time, SV1-vaccinated pullets reached maximal SE-IgA levels by 14 and 21 d p.v., while SE-IgA levels in SV2-, LPS-, or V-vaccinated pullets did not significantly change over time.

Following a secondary vaccination by i.d. GF-pulp vaccination, there was a time main effect for SE-specific IgA levels in the blood plasma of 19-week-old LBL pullets (*p* = 0.003; Figure 1). Regardless of treatment, SE-IgA levels reached maximal levels at 5 and 7 d p.v., then decreased by 14, 21, and 28 d p.v.

### 3.2. Leukocyte Profiles in GF Pulps Following Salmonella Bacterin Vaccine Administration

Temporal and quantitative changes in local GF-pulp leukocyte profiles were monitored following a primary or secondary vaccination by i.d. GF-pulp injection of SV1, SV2, LPS, or V in naïve (Group 1) and previously s.c. vaccinated (Group 2) pullets, respectively.

#### 3.2.1. Heterophils

Following the primary vaccination (Group 1) by i.d. GF-pulp injection, there was a time main effect for heterophil infiltration levels (*p* < 0.001; Figure 2). Regardless of treatment, heterophil levels sharply increased from pre-injection (0 d) to 6 h p.i., decreased from 6 h to 1 d p.i., and subsequently returned to near pre-injection levels by 2 and 3 d p.i.

Following the secondary vaccination (Group 2) by i.d. GF-pulp injection, there was a time main effect for heterophil infiltration levels (*p* < 0.001; Figure 2). Regardless of treatment, there was a stepwise increase in heterophil levels from pre-injection (0 d) to 6 h p.i. and again from 1 to 3 d p.i. followed by a stepwise decrease from 3 to 5 d p.i. and then decreased from 5 to 7 d p.i. to near pre-injection levels.

#### 3.2.2. Macrophages

Following primary vaccination by i.d. GF-pulp injection, there was a time main effect for macrophage infiltration levels (*p* = 0.006; Figure 2). Regardless of treatment, macrophage levels of injected GF pulps sharply increased from pre-injection (0 d) to 6 h p.i. and subsequently decreased from 6 h to 1 d p.i. Although numerically greater, macrophage levels returned to near pre-injection levels by 2 and 3 d p.i.

Following secondary vaccination by i.d. GF-pulp injection, there was a time main effect for macrophage infiltration levels (*p* < 0.001; Figure 2). Regardless of treatment, macrophage levels of injected GF pulps remained at pre-injection (0 d) levels until 3 d p.v. and subsequently decreased to below pre-injection levels by 5 and 7 d p.v.

#### 3.2.3. Total Lymphocytes

Following primary vaccination by i.d. GF-pulp injection, there were time (*p* = 0.006) and treatment (*p* < 0.001) main effects for total lymphocyte levels (Figure 2). Regardless of treatment, lymphocyte levels sharply increased from pre-injection (0 d) to 6 h p.i. and subsequently decreased to above pre-injection levels by 1 d p.i. For the main effect of the treatment, V-injected GF pulps had greater total lymphocyte levels than those injected with SV1, SV2, or LPS.

Following secondary vaccination by i.d. GF-pulp injection, there were treatment-by-time interactions for total lymphocyte levels (*p* < 0.001; Figure 2). GF pulps injected with V had greater lymphocyte levels than all other treatments at 6 h p.i. and greater levels than those injected with SV2 at 2 d p.i. Additionally, only GF pulps injected with V exhibited changes in lymphocyte levels over time, where their levels greatly increased from pre-injection (0 d) to 6 h p.i. and subsequently returned to near pre-injection levels by 1 d p.i.

#### 3.2.4. B Cells

Following primary vaccination by i.d. GF-pulp injection, there were time (*p* = 0.006) and treatment (*p* < 0.001) main effects for B cell infiltration levels (Figure 3). Regardless of treatment, B cell levels of injected GF pulps reached maximal levels by 3 d p.i. For the main effect of the treatment, GF pulps injected with V had greater B cell levels than all other treatments.

Following secondary vaccination by i.d. GF-pulp injection, there were treatments by time interactions for B cell infiltration (*p* < 0.001; Figure 3). There was no difference in treatments until 2 d p.i., where B cell levels of injected GF-pulps were greater in V-injected pulps when compared to those from all other treatments. At 3 d p.i., B cell levels in LPS-injected GF pulps were greater than those from all other treatments. SV2-injected GF pulps had greater B cell levels than all other treatments at 5 d p.i. while LPS-injected GF pulps had greater B cell levels than all other treatments at 7 d p.i. Over time, B cell levels from SV1- and LPS-injected GF pulps exhibited a stepwise increase from pre-injection (0 d) to 5 d p.i. and from 5 to 7 d p.i., while SV2-injected GF pulps reached maximal B cell levels at 5 d p.i. and decreased to above pre-injection levels at 7 d p.i.

#### 3.2.5. Total T Cells

Following the primary vaccination by i.d. GF-pulp injection, there were time (*p* < 0.001) and treatment (*p* < 0.001) main effects for total T-cell infiltration (Figure 3). Regardless of treatment, total T cell levels steeply increased from pre-injection (0 d) to 6 h p.i. decreased from 6 h to 1 d p.i., and ultimately returned to near pre-injection levels by 2 d p.i. For the main effect of the treatment, V-injected GF pulps had greater total T cell levels than those from all other treatments.

Following the secondary vaccination by i.d. GF-pulp injection, there were treatment by time interactions for total T cell levels (*p* < 0.001, Figure 3). GF pulps injected with V had greater total T cell levels than SV1-, SV2-, or LPS-injected GF pulps at 6 h p.i. and greater levels than SV2-injected GF pulps at 2 d p.i. Additionally, total T cell levels in V-injected GF pulps steeply increased from pre-injection (0 d) to 6 h p.i. and subsequently returned to near pre-injection levels by 1 d p.i. However, there was no change in total T cell levels over time with any of the other GF-pulp-injected treatments.

### 3.3. T Cell Subpopulations in GF Pulps Following Secondary Administration of Salmonella Bacterin Vaccines

Due to the influence of the vaccine vehicle (V) on the T cell response following primary vaccination (Group 1) by i.d. GF-pulp injection in this study (Figure 3) and by Santamaria et al. [25], analysis of T cell subpopulations (CD4^+^, CD8α^+^, and γδ T cells) in the GF pulp following a secondary vaccination (Group 2) by i.d. GF-pulp injection were assessed (Figure 4).

#### 3.3.1. CD4^+^ T Cells

Following the secondary vaccination by i.d. GF-pulp injection, there were treatment by time interactions for CD4^+^ T cell levels in injected GF pulps (*p* < 0.001; Figure 4). Levels were not different between treatments at pre-injection (0 d). At 6 h, p.i., V-injected GF-pulps had greater infiltration than all other treatments. There was no difference in CD4^+^ T cell levels between treatments at 1 d p.i. At 2 d, p.i., V-injected GF pulps had greater CD4^+^ T-cell infiltration than those injected with SV1 and SV2, but levels were not different from those injected with LPS. There was no difference in CD4^+^ T cell levels between treatments at 3, 5, and 7 d p.i. Over time, CD4^+^ T cells reached maximal levels in SV1-injected GF-pulps by 6 h p.i., did not change over time in SV2- and LPS-injected GF pulps, and reached maximal levels at 6 h and 2 d p.i in V-injected GF pulps.

#### 3.3.2. CD8α^+^ T Cells

Following the secondary vaccination by i.d. GF-pulp injection, there was a time main effect for CD8α^+^ T cell levels of injected GF-pulps (*p* < 0.001; Figure 4). Regardless of treatment, there was a stepwise decrease in CD8α^+^ T cell levels from pre-injection (0 d) levels to 1 d p.i. and again from 2 to 3 d p.i. to below pre-injection by 5 and 7 d p.i.

#### 3.3.3. γδ T Cells

Following the secondary vaccination by i.d. GF-pulp injection, there were treatment by time interactions for γδ T cell levels in injected GF pulps (*p* < 0.001; Figure 4). γδ T cell levels were not different between treatments at pre-injection (0 d). At 6 h p.i., V-injected GF-pulps had higher γδ T cell levels than all other treatments. There was no difference in γδ T cell levels between treatments at 1 d p.i. At 2 d, p.i., V-injected GF-pulps had greater γδ T cell levels than those injected with SV1 and SV2 but were not different from those injected with LPS. There was no difference in γδ T cell levels between treatments at 3, 5, and 7 d p.i. Over time, γδ T cells reached maximal levels by 6 h post V- or SV1 injection and did not change post-SV2 or -LPS injection.

### 3.4. Relative mRNA Expression of Inflammatory Cytokines in GF Pulps Following Salmonella Bacterin Vaccine Administration

Temporal and quantitative changes in relative inflammatory cytokine mRNA expression profiles (40 − ∆CT) at the site of injection were monitored following primary (Group 1) or secondary (Group 2) vaccination by i.d. administration of SV1, SV2, LPS, or V into GF-pulps (Figure 5A). There were no main effects (time or treatment) or interactions (treatment by time) for the relative mRNA expression for IL-12α or TNF-α following primary or secondary i.d. vaccinations (*p* > 0.05; Figure 5B).

#### 3.4.1. IL-1β mRNA Expression

Following the primary vaccination by i.d. GF-pulp injection, there was a time main effect for relative IL-1β expression in the injected GF pulps (*p* = 0.001; Figure 5A). Regardless of treatment, relative IL-1β expression levels sharply increased from pre-injection (0 d) to 6 h p.i. and subsequently decreased to levels above those at pre-injection from 6 h to 1 d p.i.

Following the secondary vaccination by i.d. GF-pulp injection, there was a time main effect for IL-1β expression in the injected GF-pulps (*p* < 0.001; Figure 5A). Regardless of treatment, relative IL-1β expression increased from pre-injection (0 d) to 6 h p.i., remained elevated until 5 d p.i., subsequently returning to pre-injection levels at 5 and 7 d p.i.

#### 3.4.2. IL-6 mRNA Expression

Following the primary i.d. GF-pulp injection, there was a main effect of time for relative IL-6 expression in the injected GF pulps (*p* = 0.01; Figure 5A). Regardless of treatment, relative IL-6 expression increased from pre-injection (0 d) to 6 h p.i. and subsequently decreased to levels above those at pre-injection from 1 to 3 d p.i.

Following the secondary vaccination by i.d. GF-pulp injection, there were time (*p* = 0.04) and treatment (*p* = 0.02) main effects for relative IL-6 expression in the injected GF-pulps (Figure 5A). Regardless of treatment, relative IL-6 expression increased from pre-injection (0 d) to 2 d p.i. but was not different from pre-injection or 2 d p.i. expression levels at 6 h, 1 d, 3 d, 5 d, or 7 d p.i. For the main effect of the treatment, V-injected GF-pulps had greater relative IL-6 expression compared to SV1- or LPS-injected GF-pulps but was not different from SV2-injected GF-pulps.

#### 3.4.3. IL-8 (CXCL8) mRNA Expression

Following the primary vaccination by i.d. GF-pulp injection, there was a time main effect for relative IL-8 expression levels of injected GF-pulps (*p* < 0.001; Figure 5A). Regardless of treatment, relative IL-8 expression increased from pre-injection (0 d) to 6 h p.i. and subsequently decreased to levels above those at pre-injection from 6 h to 2 d p.i.

Following the secondary vaccination by i.d. GF-pulp injection, there was a time main effect for relative IL-8 expression of injected GF-pulps (*p* < 0.001; Figure 5A). Regardless of treatment, relative IL-8 expression increased from pre-injection (0 d) to 1 d p.i. and remained elevated through 3 d p.i.

#### 3.4.4. IL-10 mRNA Expression

Following the primary vaccination by i.d. GF-pulp injection, there was a time main effect for relative IL-10 expression of injected GF-pulps (*p* = 0.002; Figure 5A). Regardless of treatment, relative IL-10 expression increased from pre-injection (0 d) to 6 h p.i., remained elevated at 1 d p.i., and subsequently decreased to pre-injection levels by 2 d p.i.

Following the secondary vaccination i.d. of the GF-pulp injection, there was a time main effect for relative IL-10 expression in injected GF-pulps (*p* = 0.050; Figure 5A). Regardless of treatment, relative IL-10 expression increased from pre-injection (0 d) to 6 h p.i. and returned to near pre-injection levels by 7 d p.i., but expression was not different from 6 h p.i. or pre-injection levels at 1, 2, 3, or 5 d p.i.

### 3.5. Relative mRNA Expression of Effector T Cell Cytokines in GF Pulps Following Salmonella Bacterin Vaccine Administration

Temporal and quantitative changes in relative T effector cell cytokine mRNA expression profiles (40 − ∆CT) were monitored following primary (Group 1) or secondary (Group 2) i.d. administration of SV1, SV2, LPS, or V into GF pulps (Figure 6A). There were no main effects (time or treatment) or interactions (treatment by time) for the relative mRNA expression of IFN-γ, IL-4, or TGF-β1 following primary and secondary i.d. GF-pulp injections (*p* > 0.05; Figure 6B).

#### 3.5.1. IL-13 mRNA Expression

Following the primary vaccination by i.d. GF-pulp injection, there were time (*p* = 0.03) and treatment (*p* = 0.003) main effects for relative IL-13 expression of injected GF pulps (Figure 6A). Regardless of treatment, relative IL-13 expression was greatest at 6 h p.i. when compared to 3 d p.i. but was not different from expression at pre-injection (0 d), 1 d, or 2 d p.i. For the main effect of the treatment, relative IL-13 expression from V-injected GF-pulps had greater expression compared to all other injected treatments.

Following the secondary vaccination by i.d. GF-pulp injection, there were treatment by time interactions for relative IL-13 expression in injected GF pulps (*p* = 0.04; Figure 6A). There was no difference in IL-13 expression between treatments at pre-injection. At 6 h p.i., SV1- and SV2-injected GF pulps had a greater IL-13 expression compared to LPS- or V-injected GF pulps. At 1 d, p.i., SV1- and V-injected GF pulps had greater IL-13 expression than LPS-injected GF pulps but were not different from SV2-injected GF pulps. There was no difference in IL-13 expression between treatments at 2, 3, or 5 d p.i. At 7 d, p.i., SV1-injected GF pulps had greater IL-13 expression than all other treatments. Over time, IL-13 expression reached maximal levels by 7 d p.i. in SV1-injected GF-pulps, did not change in SV2- or LPS-injected GF pulps, and reached maximal levels by 2 d p.i. in V-injected GF pulps.

#### 3.5.2. IL-17A mRNA Expression

Following the primary vaccination by i.d. GF-pulp injection, the treatment had a main effect for the relative IL-17A expression of injected GF pulps (*p* = 0.02; Figure 6A). IL-17A expression in SV1-injected GF-pulps was greater than in those injected with LPS, but not different from those injected with SV2 or V.

Following the secondary vaccination by i.d. GF-pulp injection, there were time (*p* = 0.046) and treatment (*p* = 0.02) main effects for relative IL-17A expression of injected GF-pulps (Figure 6A). Regardless of treatment, relative IL-17A expression reached maximal levels at 3 d p.i. when compared to pre-injection (0 d) expression levels but was not different from expression levels at 6 h, 1 d, 2 d, 5 d, or 7 d p.i. For the main effect of the treatment, IL-17A expression in V-injected GF pulps was greater than in LPS-injected GF-pulps but was not different from those injected with SV1 or SV2.

## 4. Discussion

Systemic inflammatory responses to live *Salmonella* challenges in chickens are well documented [36,37], and studies have investigated the immunological memory of *Salmonella* vaccine candidates [32,38]. However, there is limited knowledge of the quantitative and temporal characteristics of systemic humoral (blood) and local cellular (GF pulp) immune responses in egg-type pullets after vaccination with autogenous *Salmonella* bacterin vaccines suspended in a water–oil–water emulsion.

### 4.1. Vaccination with Autogenous Salmonella Bacterin Induced a Humoral Response with Evidence of Immunological Memory

IgM is one of the first immunoglobulins to respond to an infection and neutralize pathogenic bacteria [39]. In this study, SE-specific IgM levels in blood plasma were responsive to vaccination with autogenous *Salmonella* bacterin. Pullets s.c.-vaccinated and i.d.-injected (Group 2) with SV1 or SV2 had nearly two times greater SE-specific IgM levels than the V-control treatment by 10 d post-vaccination. Similar results were obtained by Santamaria et al. [25] when *Salmonella* bacterin vaccines formulated in a water–oil–water emulsion (the same vaccine formulations used in the present study) were i.d. injected into GF-pulps. Specifically, a primary i.d. GF-pulp injection with SV1 or SV2 in LBL pullets resulted in maximal SE-specific IgM levels in blood plasma by 7 d p.v. which remained elevated through 28 d p.v. [25]. The results of this study and Santamaria et al. [25] also resemble previous studies that assessed SE-specific IgM after primary exposure to live *Salmonella* in broiler chicks [37]. Additionally, in the present study, secondary vaccination by i.d. GF-pulp injection with LPS alone produced a transient IgM peak from 3 to 10 d p.v. This transient peak was also observed by Santamaria et al. [25] following a primary i.d.-LPS injection beginning at 5 d p.v. and resolving by 10 d p.v. Transient peaks in IgM are common markers of a humoral response to LPS and have been observed in chickens, other animal models, and during in vitro investigations [5,20,30,31,32].

The present study showed a gradual quantitative shift from SE-specific IgM to IgG during the first week of post-primary s.c. vaccination, which was heightened and maintained after the secondary i.d. GF-pulp injection with SV1 or SV2. Following a primary vaccination by i.d. GF-pulp vaccination with *Salmonella* bacterin, Santamaria et al. [25] also observed a similar quantitative shift from SE-specific IgM to IgG from 5 d to 14 d p.v. These qualitative, quantitative, and temporal changes in the systemic humoral response to SV1 and SV2 are indicative of a T-dependent response, with T helper cells driving the isotype switch and immunological memory development, ultimately leading to an effective humoral response [40].

Another antibody isotype is IgA, which is a secretory immunoglobulin present in mucosal secretions like saliva, bile, mucus, albumen, and feces, but it can also be found in circulating blood [39]. Poultry studies detected increased IgA levels in the lumen of the gastrointestinal tract in the days following an oral challenge with pathogenic *Salmonella* serovars [6,36], but *Salmonella*-specific IgA levels in peripheral blood are generally much lower than IgM or IgG after *Salmonella* vaccination or challenge [25,30]. In this study, SE-specific IgA levels in peripheral blood plasma were lower than SE-specific IgM or IgG. Even so, primary s.c. vaccination with SV1 resulted in elevated SE-specific IgA plasma levels by 14 d post-vaccination which was maintained through the first 10 d post-secondary vaccination by i.d. GF-pulp injection. Similarly, Marouf et al. [41] observed a prolonged elevation of SE-specific IgA levels in the blood of specific pathogen-free Rhode Island chicks orally vaccinated thrice with a trivalent *Salmonella* outer membrane vesicle vaccine. However, when assessing SE-specific antibody levels in blood plasma after primary i.d. GF-pulp injection with *Salmonella* bacterin formulated in a water–oil–water emulsion, Santamaria et al. [25] found that relative levels of SE-specific IgA reached maximal levels by 7 d post-i.d. GF-pulp injection with SV1 or SV2 and decreased to levels above those at pre-injection by 21 d p.v. This SE-specific IgA response observed by Santamaria et al. [25] was one week faster than the temporal progression of the present study, which may be because their primary vaccination was administered by i.d. GF-pulp injection, while a primary vaccination was administered by s.c. administration in this study. Even so, like the present study, Santamaria et al. [25] observed lower SE-specific IgA levels in blood plasma compared to SE-specific IgM or IgG levels.

The SE-specific antibody profiles of this study differ from Beal et al. [42] and Withanage et al. [43]. For example, Beal et al. [42] observed that a secondary challenge with live *S*. Typhimurium (ST) in 16-week-old specific-pathogen-free White Leghorns revealed numerically lower *Salmonella* protein antigen (STAgP)-specific IgM levels when compared to levels following their primary *Salmonella* challenge or to an age-matched primary *Salmonella* challenge. While STAgP-specific IgM and IgG levels remained elevated weeks after the primary challenge, there was no further increase following a secondary challenge [42]. Withanage et al. [43] observed similar results to Beal et al. [42], where a secondary challenge with live ST in 10-week-old specific-pathogen-free Rhode Island Red chickens resulted in no changes to ST protein antigen-specific IgM and IgG in peripheral blood. However, they observed an increase in antigen-specific IgA in the blood at 14 d post-secondary challenge. Differences in the results of Beal et al. [42] and Withanage et al. [43] when compared to the present killed SV1 and SV2 study may be attributed to their use of live ST challenges. Additionally, these studies [28,29] quantified protein-antigen-specific antibody responses while the present study monitored the antibody response to formalin-killed, whole SE bacteria, likely generating a more diverse repertoire of antibody specificities.

Moreover, the present study further validates the quantitative, qualitative, and temporal characteristics of the SE-specific, primary antibody responses to SV1 or SV2 in age-matched LBL pullets reported by Santamaria et al. [25], although they administered the vaccines by i.d. GF-pulp injection in contrast to the s.c. vaccination used in the current study. Even so, evidence suggests that commercial *Salmonella* vaccination programs maintain elevated *Salmonella*-specific antibody levels in peripheral blood above pre-vaccination through 60 days post-vaccination [44], but the waning period of humoral protection is not well defined for these vaccination programs and varies widely based on bird type, vaccine, and study [10,41,45,46,47].

### 4.2. Secondary Intradermal Injection of Autogenous Salmonella Bacterin Stimulated Prolonged Heterophil and Macrophage Activity at the Site of Injection

Heterophils, the avian counterpart to mammalian neutrophils, are one of the first leukocytes to localize at the site of infection. At the site of infection, these cells become highly efficient phagocytes and produce oxidative bursts, secrete antimicrobial peptides, and synthesize extracellular traps to remove and destroy invading microbes [48,49,50]. In this study, a primary vaccination by i.d. GF-pulp injection with autogenous *Salmonella* bacterin vaccines or vaccine components exhibited an influx in heterophils at 6 h p.i. followed by a drop to pre-injection levels by 2 d p.i. This differed from the heterophil infiltration observed following secondary vaccination by i.d. GF-pulp injection with these vaccine treatments in the previously s.c. immunized chickens. Specifically following the secondary i.d. injection, the local heterophil levels remained elevated through 5 d p.i. Santamaria et al. [25] also observed maximal heterophil levels by 6 h p.i. following a primary vaccination with the same *Salmonella* bacterin vaccines i.d. GF-pulp injection, emphasizing the replicability of the GF-pulp injection model when assessing cellular responses to *Salmonella* vaccines in poultry.

Like the longitudinal changes in local heterophil levels after *Salmonella* bacterin injection, there were differing temporal characteristics in macrophage levels following primary or secondary vaccination by i.d. GF-pulp injections with *Salmonella* bacterin vaccines or vaccine components. Regardless of vaccine treatment, maximal macrophage levels were reached by 6 h post-primary i.d. GF-pulp injection, while a diphasic increase in macrophage levels was observed at 6 h and 3 d post-secondary i.d. GF-pulp injection. However, after 3 d post-secondary i.d., macrophage levels fall below those at pre-injection, which may be due to these phagocytes leaving the site of injection to travel to secondary lymphoid organs to facilitate antigen presentation to lymphocytes [51]. Even so, these results resembled past studies which assessed local macrophage levels following i.d. GF-pulp injections with bacterial antigens. Specifically, Santamaria et al. [25] determined that primary vaccination with *Salmonella* bacterin vaccines by i.d. GF injection resulted in sustained, elevated macrophage levels from 6 h to 2 d post-GF-pulp injection, while French et al. [28] observed that primary i.d. GF-pulp injection with ST-LPS resulted in a stepwise-increase in macrophage levels from 0 h to 6 h then from 6 h to 24 h p.i. in 34 d old broilers.

During an inflammatory response, IL-8 (chemokine CXCL8/CXCLi2 [52]) is produced by macrophages, neutrophils (heterophils in avian species), endothelial cells, and epithelial cells to initiate and maintain neutrophil chemotaxis in murine models [53]. Likewise, IL-8 stimulates heterophil chemotaxis in poultry models [54]. For example, an IL-8 antagonist reduced heterophil presence in the abdominal cavity of Leghorn chicks after intra-abdominal injection with live SE [54], and in vitro investigations discovered that heterophils are responsive to *Salmonella* flagellin and upregulate IL-8 expression [55]. In this study, the effect of relative IL-8 mRNA expression on heterophil chemotaxis was evident since IL-8 expression and heterophil presence reached their greatest levels by 6 h post-primary i.d. GF-pulp injection and by 2- and 3 d post-secondary i.d. GF-pulp injection. Like the results of this study, French et al. [28] determined that a primary i.d. GF-pulp injection with ST-LPS resulted in elevated relative IL-8 mRNA expression at 6 h post-i.d. injection.

Along with the IL-8 chemokine, the relative mRNA expressions of IL-1β and IL-6 inflammatory cytokines were also measured. Relative IL-1β mRNA expression remained above pre-injection levels from 6 h to 3 d post-primary and -secondary vaccination by GF-pulp injection, regardless of treatment, indicating the innate inflammatory activity of these cytokines. The temporal changes in IL-6 expression closely resembled that of IL-8 expression, with maximal IL-6 levels observed at 6 h and 1 d post-primary i.d. GF-pulp injection and at 2 d post-secondary i.d. GF-pulp injection. Similarly, French et al. [28] found that primary i.d. GF-pulp injection with ST-LPS in 34 d old broiler chicks stimulated an increase in local relative IL-1β mRNA expression from 0 h to 6 h p.i. followed by a decrease to above-pre-injection levels at 24 h p.i., meanwhile relative IL-6 mRNA expression gradually increased from 0 h to 24 h p.i. In an in vitro study that investigated immunostimulatory effects of ST outer membrane vesicles in chicken macrophage HD11 cell cultures, Cui et al. [56] determined that these ST vesicles as well as LPS alone stimulated dendrite formation and increased relative mRNA expression of LITAF, IL-6, and iNOS in vitro. However, the neutralization of LPS in the outer membrane vesicles resulted in lower IL-1β, LITAF, IL-6, and IL-10 expression in HD11 cells [56]. Therefore, many of the acute inflammatory responses observed in live animal models and by in vitro assays may be largely stimulated by LPS rather than surface markers expressed on a whole bacterium.

Functionally, IL-1β and IL-6 are pro-inflammatory cytokines produced by many cell types (including macrophages), and both play key roles in the acute phase response and in CD4^+^ T helper (Th) 17 differentiation, inhibiting CD4^+^ T regulatory cell differentiation, activating CD8^+^ cytotoxic T cell differentiation, and activating antibody production by B cells [57,58]. The functional influences of these cytokines are evident when evaluating local T cell subpopulations and relative mRNA expression of effector T cell responses following primary and secondary vaccination by i.d. GF-pulp injection. Specifically, secondary i.d. GF-pulp injection produced higher γδ T cell levels at 2 d p.i. in SV1-injected GF-pulps along with greater relative IL-13 mRNA expression (Th2-type response) at 6 h, 1 d, and 7 d p.i. and elevated relative IL-17A mRNA expression (Th17-type response) through 3 d post-secondary i.d. GF-pulp injection. IL-17A is associated with neutrophil recruitment to the site of inflammation in murine models [59], which corresponds with the elevated expression of heterophil chemotaxis cytokines (IL-17A and IL-8) and sustained heterophil presence at 2 and 3 d post-secondary i.d. GF-pulp injection in this study. Although T cell subpopulations were not assessed after the primary i.d. GF-pulp injection in this study, it is important to note that maximal IL-13 mRNA expression occurred at 6 h post-primary i.d. GF-pulp injection, which corresponds with the elevated primary innate leukocyte and total lymphocyte cell infiltration profiles at 6 h p.i. Temporal characteristics of cell-mediated immunity at the site of injection will be further discussed in the next section.

To regulate an acute inflammatory response, the anti-inflammatory mediator cytokine IL-10 is produced by most leukocytes, including macrophages and T lymphocytes [60]. Following a primary- or secondary-vaccination by i.d. GF-pulp injection in this study, relative IL-10 mRNA expression of vaccine-injected GF-pulps reached maximal levels at 6 h p.i. and returned to near pre-injection levels by 2 d p.i. Similar results were observed when GF-pulps of 34 d old broilers were i.d.-injected with ST-LPS, where IL-10 mRNA expression peaked at 6 h p.i. then decreased to above pre-injection levels at 24 h p.i. [28]. Interestingly, even though there was a similar time course for relative IL-10 mRNA expression following a primary- or secondary-i.d. GF-pulp injection with *Salmonella* bacterin vaccines or vaccine components, a secondary vaccination by i.d. GF-pulp injection still produced an extended local inflammatory response as indicated by elevated and sustained heterophil and macrophage levels along with elevated relative IL-6 and IL-8 mRNA expression.

Ultimately, a secondary vaccination with autogenous *Salmonella* bacterins by injection into the GF-pulps maintained a sustained local heterophil and macrophage presence in conjunction with elevated IL-6, IL-8, and IL-17A mRNA expression. Following a second administration with *Salmonella* bacterins, these prolonged innate immune responses suggest some form of local, chronic inflammation involving an adaptive inflammatory Th17 mechanism. These results could also be evidence of a type of trained innate immunity, but this is not well described in poultry, and further investigations are necessary to understand the significance of these prolonged inflammatory responses for disease resolution [61,62,63].

### 4.3. A Secondary Intradermal Vaccination of Autogenous Salmonella Bacterin Stimulated Cell-Mediated Activities at the Site of Injection

CD8^+^ T cells are commonly termed “cytotoxic T cells” that respond to intracellular pathogens and tumors, but there are subpopulations of CD8^+^ cells (γδ T cells, natural killer cells, dendritic cells, innate lymphoid cells) that generate a variety of cellular activities that have been identified in poultry [64]. In this study, secondary i.d. GF-pulp injection found that the total CD8α^+^ T cell population began to decrease (regardless of vaccine treatment) at 1 d p.i. and was below pre-injection level by 3 d p.i. Similarly, pullets s.c.-vaccinated with killed SE bacterin had lower CD8^+^ T cell levels in the liver by 1 d p.i. and lower CD8^+^ T cell levels in the cecal tonsils by 6 d p.i., indicating a systemic reduction in the CD8^+^ T cell response to *Salmonella* bacterin [6]. *Salmonella* is an intracellular pathogen, and it was expected that administration of a *Salmonella* vaccine would stimulate a CD8α^+^ T cell response with associated IFN-γ mRNA expression in this study. For example, Carvajal et al. [36] observed increased CD8^+^ T cell localization in the ceca and an increased ratio of CD8^+^/γδTCR^+^ lymphocytes in the blood at 6 d post-live SE vaccination. In the present study, and the study by Penha Filho et al. [6], the *Salmonella* vaccines were killed bacteria which may not initiate a cytotoxic T cell response like that observed by Carvajal et al. [36].

In this study, secondary vaccination by i.d. GF-pulp vaccination with SV1 resulted in greater levels of γδ T cells at 2 d p.i. than those observed with SV2 injection. The autogenous SV1 bacterin differs from SV2 in endotoxin unit concentration; SV1 contains 180,000 endotoxin units/dose while SV2 contains 68,000 endotoxin units/dose. Therein, this suggests that the LPS concentration in these vaccines could influence the type of T cell response. Additionally, the increase in γδ T cell infiltration at 2 d p.i. also corresponds with elevated IL-1β and IL-17A expression at 2 and 3 d p.i. in SV1-injected GF-pulp, respectively, indicating a preference for Th17-type responses to *Salmonella* bacterin with high endotoxin dosage. However, it is possible that the different mixtures of *Salmonella* serovars in SV1 and SV2 may influence the type of Th cell responses stimulated [65]. For example, in vitro infection of chicken monocyte-derived macrophages and CD4^+^ T cells with live SE resulted in higher IL-17F, TGF-β4, IL-6, CXCLi2, IL-12α, IL-18, and IL-10 when compared to cells infected with live *S.* Pullorum, demonstrating that SE preferentially stimulated a Th2/Th17-type response [65].

Additionally, regardless of vaccine treatments, B cell levels gradually increased through 3 d post-primary vaccination by i.d. GF-pulp injection, while levels increased by 5 and 7 d post-secondary i.d. GF-pulp injection. Comparatively, extended B cell infiltration was observed in 1 d old broiler chicks after oral challenge with SE or *S.* Infantis, where elevated B cell levels were still evident in the cecal tonsils by 4 d post-challenge [66]. The B cells identified in the present study may be associated with dispersed lymphoid tissue in the bird’s skin, and it is possible that they are directly associated with the IgM-to-IgG isotype switch observed in peripheral blood starting at 5 d post-secondary i.d. injection [67,68].

### 4.4. Water–Oil–Water Emulsion Alone Recruited High Levels of CD4^+^ and γδ T Cells at 6 h Post-Intradermal Injection

Advancements in vaccine emulsion formulations allow for the targeting of specific arms of the immune system, which ultimately push towards Th1- or Th2-cell-mediated responses [69]. However, recent studies have also determined that certain vaccine adjuvant-emulsion formulations can result in Th1/Th17- or Th2/Th17-type responses in murine models [70,71]. The incidence of a Th17 response is generally observed with Freund’s Complete Adjuvant, but the use of this emulsion in human and animal medicine has been discontinued due to its highly inflammatory nature [69]. In food animal production, many vaccines are formulated with water-in-oil emulsions (using mineral oil or vegetable oil) due to their nonspecific immunostimulatory capabilities and cost-effectiveness [72].

In the present study, a proprietary vaccine vehicle formulation (water–oil–water emulsion) recruited greater levels of lymphocytes (total lymphocytes, B cells, and total T cells) to the site of administration after primary- and secondary-vaccination by i.d. GF-pulp injection. Specifically, total lymphocytes and total T cells in V-injected GF-pulps reached maximal levels by 6 h p.i. while B cells reached maximal levels by 3 d post-primary injection and 7 d post-secondary injection. Similar quantitative and temporal changes in T cell and B cell levels following primary i.d. GF-pulp injection with the same water–oil–water emulsion were also observed by Santamaria et al. [25]. After secondary i.d. GF-pulp injection with *Salmonella* vaccines or vaccine components, additional analyses of lymphocyte subpopulations revealed that CD4^+^ T cell and γδ T cell levels at 6 h and CD4^+^ T cell levels at 2 d p.i. were greater in V-injected GF-pulps than in those injected with SV1, SV2, or LPS. CD4^+^ T helper cells are essential for activating cellular- and humoral-immune response as well as developing immunological memory.

It seems that this water–oil–water emulsion specifically targets CD4^+^ T cell recruitment, but the inclusion of the *Salmonella* bacterin vaccines may suppress the local CD4^+^ T cell response. Even so, systemic cell-mediated activities were still observed in this study since there was clear evidence of T cell-dependent B cell activation (IgM-to-IgG antibody isotype switch). These “T cell/B cell” interactions likely occurred in the spleen of *Salmonella*-vaccinated birds, especially since avian species do not have lymph nodes. For T cell-dependent B cell activation to occur, the *Salmonella* bacterin is phagocytized by antigen-presenting cells (dendritic cells and macrophages) at the site of injection where these cells degrade the *Salmonella* vaccine into antigen peptides. Next, these activated phagocytes travel to the marginal zone of the spleen and present the processed peptide in association with the major histocompatibility complex II (MHCII) to the T cell receptor of a Th cells residing in the spleen’s periarteriolar lymphoid sheath. At the same time, B cells in splenic follicles become activated when antigen-presenting cells present unprocessed *Salmonella* protein antigens to B cell receptors expressed by B cells. The B cell endocytoses the receptor-associated protein, degrades the protein into an antigen peptide, and presents the peptide in association with MHCII. These activated T and B cells travel to the margins of their respective regions in the spleen where the TCR of a Th cell can interact with MHCII-associated peptides presented on the B cell. From this interaction, a cascade of signals are produced which ultimately pushes the B cell toward antigen specificity through proliferation, antibody isotype switching, and affinity maturation [51].

In addition, the assessment of local T effector cell activities in V-injected GF-pulps revealed that relative IL-6 and IL-17A mRNA expressions were elevated. Specifically, relative IL-17A mRNA expression remained elevated above pre-injection levels through 3 d post-secondary i.d. GF-pulp injection, regardless of treatment. These results, when considered along with elevated γδ T cell levels at 6 h and 2 d post-secondary i.d. GF-pulp injection with V, suggest that water–oil–water emulsions stimulate a Th17-type response [65]. As previously stated, this prolonged heterophil- and Th17-type response may also indicate a chronic inflammation, but more studies are needed to understand how water–oil–water emulsions stimulate a pullet’s innate and adaptive immune systems, and how these responses aid or hinder immune responses when these emulsions are used to suspend *Salmonella* bacterin vaccines.

These results differ from those of another study that reported that an ST vaccine formulated in oil emulsion induced a Th1-type response (elevated relative expression of IFN-γ and IL-12) in Brown Nick pullets [73]. Considering the differing results, many factors appear to influence the type of T helper response stimulated by *Salmonella* vaccines. Factors may include, but are not limited to, poultry flock genetics, vaccine emulsions and adjuvants, and the nature of the *Salmonella* antigen (live, killed, sonicated, genetically engineered).

## 5. Conclusions

Efforts are ongoing to develop effective *Salmonella* vaccination programs for broiler breeders and table egg flocks that reduce *Salmonella* presence in birds, initiate long-term immunological memory, and do not cause severe injection site reactions. The present study is one of the first to investigate local cellular (GF-pulp, injection site) and systemic-humoral (blood plasma) responses after vaccination with autogenous *Salmonella* bacterin vaccines in a water–oil–water emulsion. Here, we provide evidence that secondary vaccinations with commercial *Salmonella* bacterin vaccines formulated in a water–oil–water emulsion can prolong heterophil recruitment to the site of injection, stimulate local Th17 cell-mediated responses, and initiate systemic humoral immunity through a Th cell-mediated IgM-to-IgG antibody isotype switch. From these results and the results of past research, it is apparent that further investigations are needed to appropriately interpret the cellular interactions stimulated locally (at the site of injection) and systemically (blood and secondary lymphoid organs) by various *Salmonella* serovars and vaccine adjuvants administered to commercial poultry flocks. Nonetheless, the combination of T cell-dependent antibody production and prolonged heterophil recruitment after vaccination with *Salmonella* bacterin formulated in a water–oil–water emulsion could be a desirable immunological response that can prevent *Salmonella* colonization in chickens.

## Figures and Tables

**Figure 1 vaccines-13-00311-f001:**
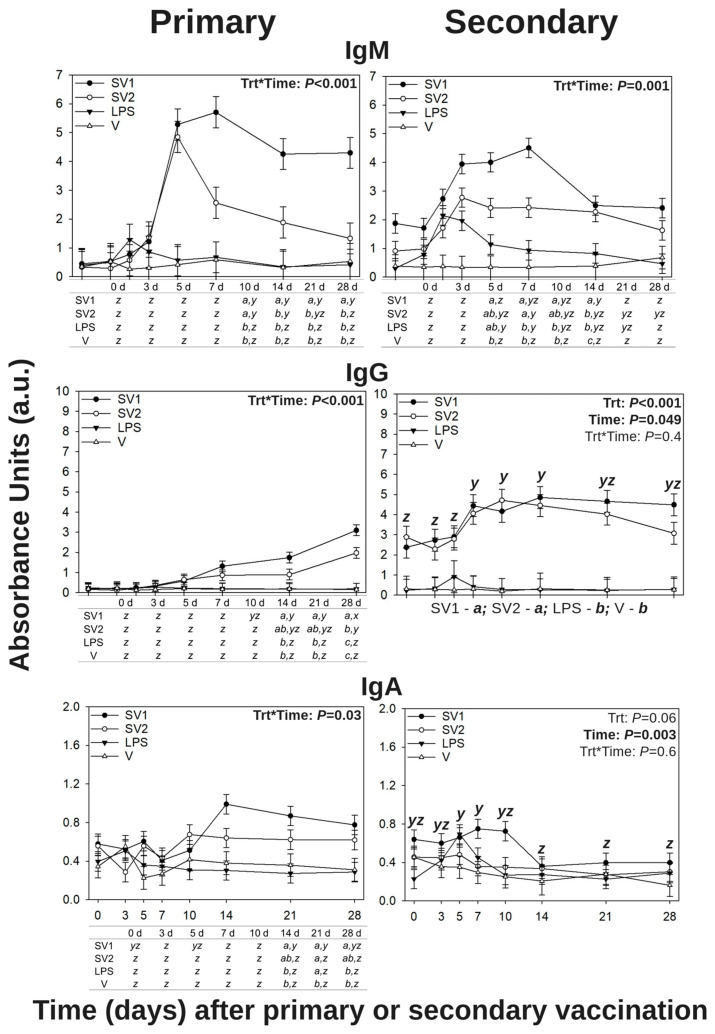
*Salmonella* Enteritidis-specific IgM, IgG, and IgA levels in the peripheral blood of Light-Brown Leghorn (LBL) pullets following a primary subcutaneous (s.c.) and secondary intradermal (i.d.) vaccination with autogenous *Salmonella* bacterin vaccines or vaccine components. *Salmonella* Enteritidis (SE)-specific antibodies in peripheral blood were measured in LBL pullets from Group 2. For the primary vaccination, 14-week-old pullets were s.c. vaccinated with 0.5 mL of autogenous *Salmonella* bacterin vaccine (SV) 1 (*n* = 4), SV2 (*n* = 4), SE lipopolysaccharide (LPS; *n* = 4), or water–oil–water emulsion vaccine vehicle (V; *n* = 3) in the nape of the neck. For the secondary vaccination, the respective s.c. treatments (trts) were i.d.-injected into the growing feather (GF) pulp at 19 weeks of age (21 GF/pullet, 10 μL/GF, 210 μL total injection volume). Heparinized blood was collected from the brachial wing vein before (0 d) and at 3, 5, 7, 10, 14, 21, and 28 d post-primary and secondary vaccination. Antibody levels were quantified by ELISA. Data were analyzed by two-way repeated measures ANOVA (trt, time, and their interactions) and are presented as mean absorbance units (a.u.) ± SEM. *x*–*z*: Timepoints that do not share the same superscript are statistically different; *a*–*c*: Trt that do not share the same superscript are statistically different; Trt*time interactions are denoted above each bar in the figure where timepoints within a trt that do not share the same superscript (*x*–*z*) are statistically different, and trt groups within a timepoint that do not share the same superscript (*a*–*c*) are statistically different.

**Figure 2 vaccines-13-00311-f002:**
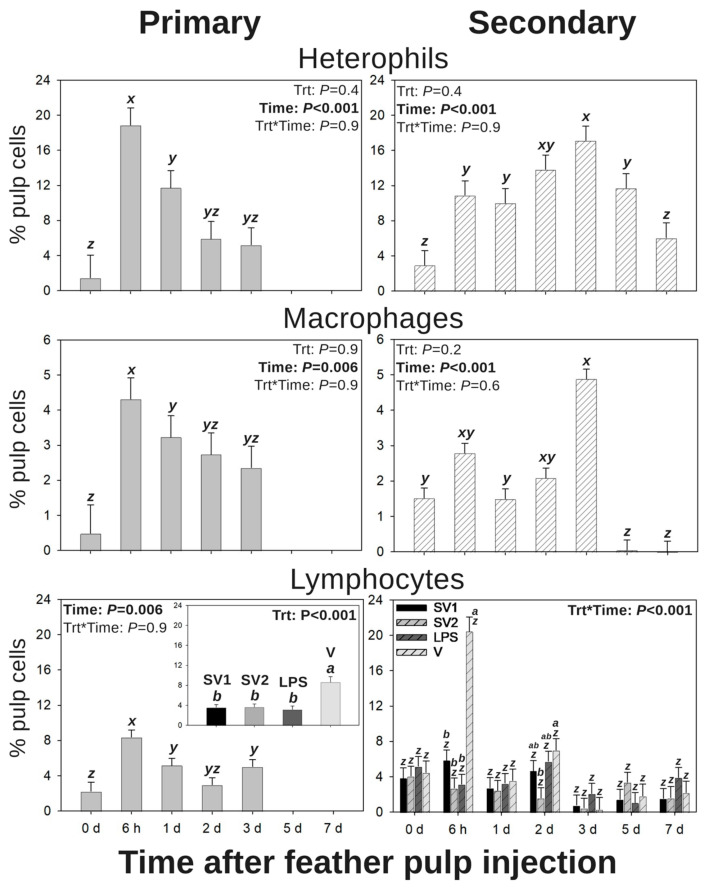
Leukocyte profiles in growing feather (GF) pulps following primary or secondary intradermal injection of commercial *Salmonella* bacterin vaccines or vaccine components in Light-Brown Leghorn (LBL) pullets. For the primary intradermal (i.d.) vaccination (Group 1), 14-week-old pullets were i.d.-injected into the GF pulp with *Salmonella* vaccine treatments (trt): SV1 (*n* = 4), SV2 (*n* = 4), SE lipopolysaccharide (LPS; *n* = 4), or water–oil–water emulsion vaccine vehicle (V; *n* = 3; 10 μL of injected/GF, 12 GF/pullet, 120 μL of total injection volume). For the secondary i.d. vaccination (Group 2), 14-week-old pullets were subcutaneously (s.c.) administered with Group 1 trt in the nape of the neck (0.5 mL), followed by i.d. GF-pulp injection at 19 wk with their respective s.c. trt (21 GF/pullet, 10 μL/GF, and 210 μL of total injection volume). Injected GF from primary and secondary i.d vaccinations were collected before (0 d) and at 6 h, 1 d, 2 d, 3 d, 5 d, and 7 d post-injection. Direct-immunofluorescent staining of GF-pulp cell suspensions with fluorescence-conjugated, chicken leukocyte-specific, mouse monoclonal antibodies (Southern Biotech, Birmingham, AL, USA) identified macrophage and lymphocyte populations. Cell population analysis was conducted using fluorescence-based flow cytometry. Heterophil gating was based on the size and granularity of CD45^+^ cells [33]. Lymphocytes were calculated by adding the percentages of CD4^+^ T cells, CD8α^+^ T cells, CD8α^−^ γδ T cells, and B cells. Data were analyzed by two-way ANOVA (trt, time, and their interactions), and shown as mean % pulp cells ± SEM. *x*–*z*: Timepoints that do not share the same superscript are statistically different; *a,b*: Trts that do not share the same superscript are statistically different; Trt*time interactions are denoted above each bar in the figure where timepoints within a trt that do not share the same superscript (*x*–*z*) are statistically different, and trt groups within a timepoint that do not share the same superscript (*a*,*b*) are statistically different.

**Figure 3 vaccines-13-00311-f003:**
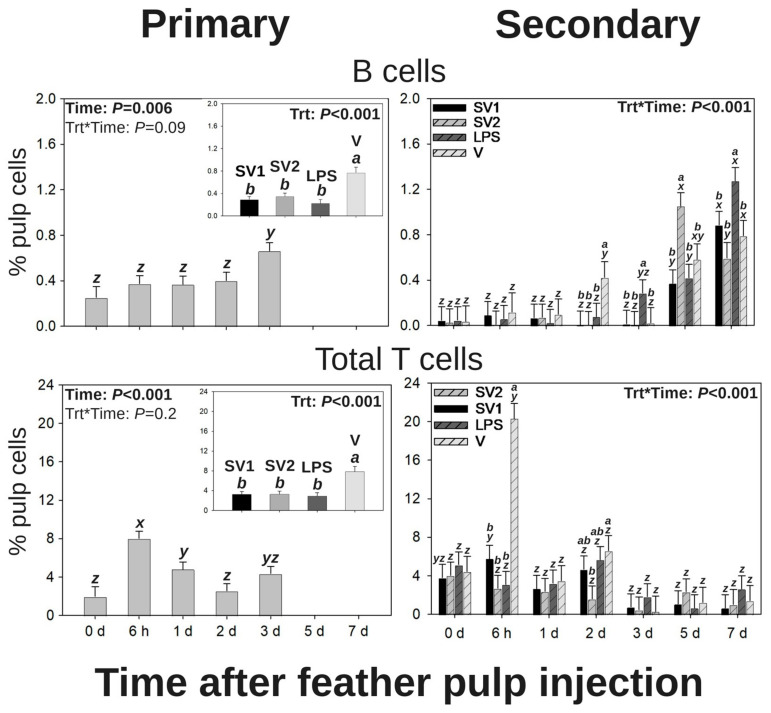
Lymphocyte profiles in growing feather (GF) pulps following primary or secondary intradermal injection of commercial *Salmonella* bacterin vaccines or vaccine components in Light-Brown Leghorn (LBL) pullets. For the primary intradermal injection, 14-week-old pullets of Group 1 were intradermally (i.d.) injected into the GF pulp with *Salmonella* vaccine treatments (trt): SV1, SV2, SE lipopolysaccharide (LPS), or water–oil–water emulsion vaccine vehicle (V; 10 μL of injected/GF, 24 GF/pullet and 240 μL of total injection volume). For the secondary intradermal injection, 14-week-old pullets of Group 2 were subcutaneously (s.c.) administered (0.5 mL) with the same treatments (trts) as Group 1 in the nape of the neck followed by i.d. GF-pulp injection at 19 weeks of age with their respective s.c. trt (21 GF/pullet, 10 μL/GF, and 210 μL of total injection volume). Injected GF from primary and secondary i.d. vaccinations were collected before (0 d) and at 6 h, 1 d, 2 d, 3 d, 5 d, and 7 d post-injection (p.i.). Direct-immunofluorescent staining of GF-pulp cell suspensions with fluorescence-conjugated, chicken leukocyte-specific, mouse monoclonal antibodies (Southern Biotech, Birmingham, AL) identified T cell and B cell populations, and cell population analysis was conducted by fluorescence-based flow cytometry. Total T lymphocytes were quantified by adding together the percentages of CD4^+^ T cells, CD8α^+^ T cells, and CD8α^−^ γδ T cells. Data were analyzed by two-way ANOVA (trt, time, and their interactions), and data are shown as mean % pulp cells ± SEM. *x–z*: Timepoints that do not share the same superscript are statistically different; *a*,*b*: Trt that do not share the same superscript are statistically different; Trt*time interactions are denoted above each bar in the figure where timepoints within a trt that do not share the same superscript (*x–z*) are statistically different, and trt groups within a timepoint that do not share the same superscript (*a*,*b*) are statistically different.

**Figure 4 vaccines-13-00311-f004:**
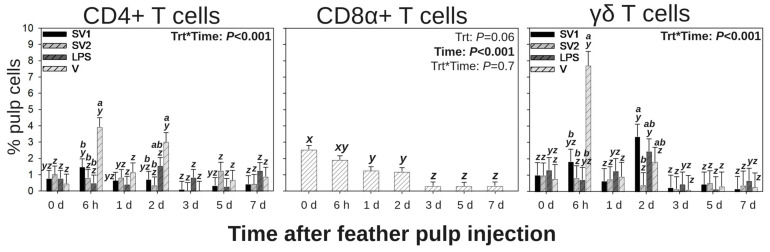
T lymphocyte subpopulations in growing feather (GF) pulps following secondary intradermal (i.d.) injection of commercial *Salmonella* bacterin vaccines or vaccine components in Light-Brown Leghorn (LBL) pullets. For the secondary intradermal injection (Group 2), 14-week-old pullets were subcutaneously (s.c.) administered (0.5 mL) with *Salmonella* vaccine treatments (trt): SV1, SV2, SE lipopolysaccharide (LPS), or water–oil–water emulsion vaccine vehicle (V) in the nape of the neck followed by secondary i.d. GF-pulp injection at 19 weeks of age with their respective s.c. trt (21 GF/pullet, 10 μL/GF, and 210 μL of total injection volume). Injected GF from primary and secondary i.d. vaccinations were collected before (0 d) and at 6 h, 1 d, 2 d, 3 d, 5 d, and 7 d post-injection (p.i.). Direct-immunofluorescent staining of GF-pulp cell suspensions with fluorescence-conjugated, chicken leukocyte-specific, mouse monoclonal antibodies (Southern Biotech, Birmingham, AL, USA) identified T cell subpopulations (CD4^+^, CD8α^+^, and total γδ T cells). Cell populations were analyzed by flow cytometry. Data were analyzed by two-way ANOVA (trt, time, and their interactions), and data are shown as mean % pulp cells ± SEM for each leukocyte population. *x–z*: Timepoints that do not share the same superscript are statistically different; *a*,*b*: Trts that do not share the same superscript are statistically different; Trt*time interactions are denoted above each bar in the figure where timepoints within a trt that do not share the same superscript (*x–z*) are statistically different, and trt groups within a timepoint that do not share the same superscript (*a*,*b*) are statistically different.

**Figure 5 vaccines-13-00311-f005:**
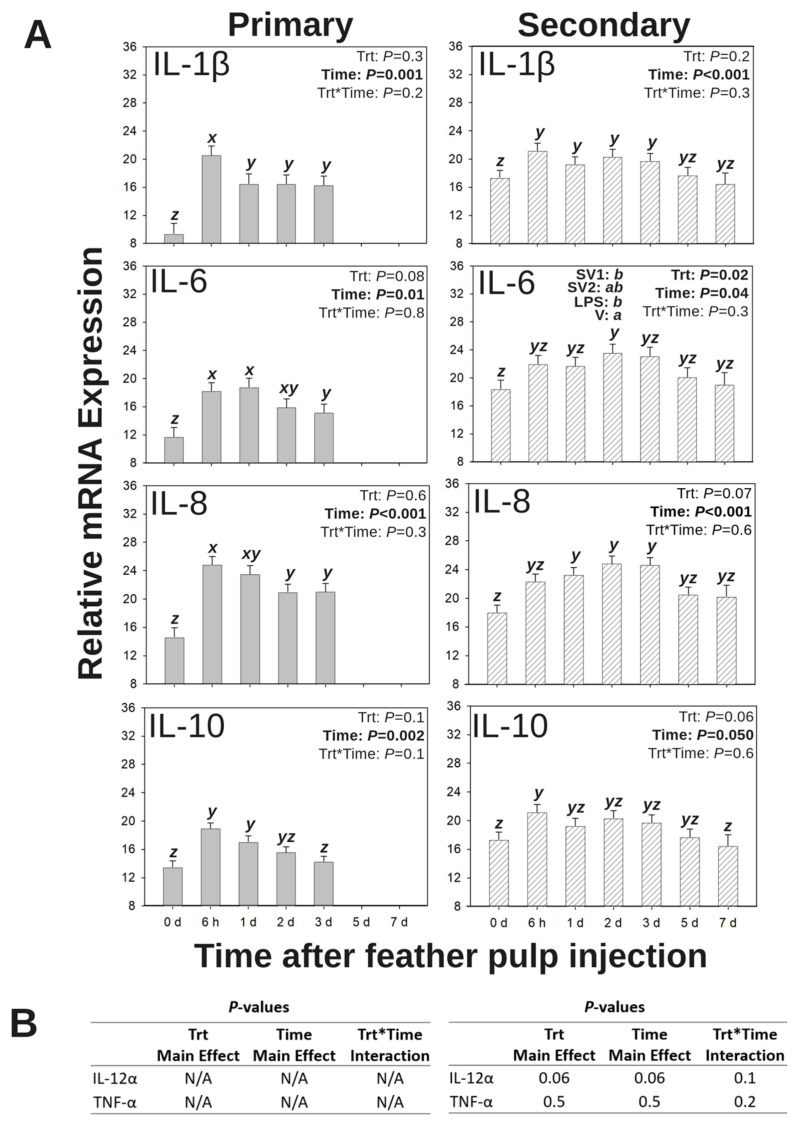
Inflammatory cytokine mRNA expression profiles in growing feather (GF) pulps following primary or secondary intradermal (i.d.) injection of commercial *Salmonella* bacterin vaccines or vaccine components in Light-Brown Leghorn (LBL) pullets. For the primary i.d. injection (Group 1), 14-week-old pullets were i.d. injected into the GF pulp with *Salmonella* vaccine treatments (trt): SV1, SV2, SE lipopolysaccharide (LPS), or water–oil–water emulsion vaccine vehicle (V; 10 μL of injected/GF, 24 GF/pullet, 240 μL of total injection volume). For the secondary i.d. injection (Group 2), 14-week-old pullets were subcutaneously (s.c.) administered (500 μL) with the same treatments trt as Group 1 in the nape of the neck followed by i.d. GF-pulp injection at 19 weeks with their respective s.c. trt (21 GF/pullet, 10 μL/GF, 210 μL of total injection volume). Injected GF from primary and secondary i.d. vaccinations were collected before (0 d) and at 6 h, 1 d, 2 d, 3 d, 5 d, and 7 d-post-injection (p.i.). Cytokine mRNA expression was measured by quantitative RT-qPCR. Data were analyzed by two-way ANOVA (trt, time, and their interactions), and presented as mean 40 − ∆C_T_ ± SEM for (**A**) IL-1β, IL-6, IL-8, and IL-10, and (**B**) IL-12α and TNF-α. *x–z*: Timepoints that do not share the same superscript are statistically different; *a*,*b*: Trts that do not share the same superscript are statistically different; Trt*time interactions are denoted above each bar in the figure where timepoints within a trt that do not share the same superscript (*x–z*) are statistically different, and trt groups within a timepoint that do not share the same superscript (*a*,*b*) are statistically different.

**Figure 6 vaccines-13-00311-f006:**
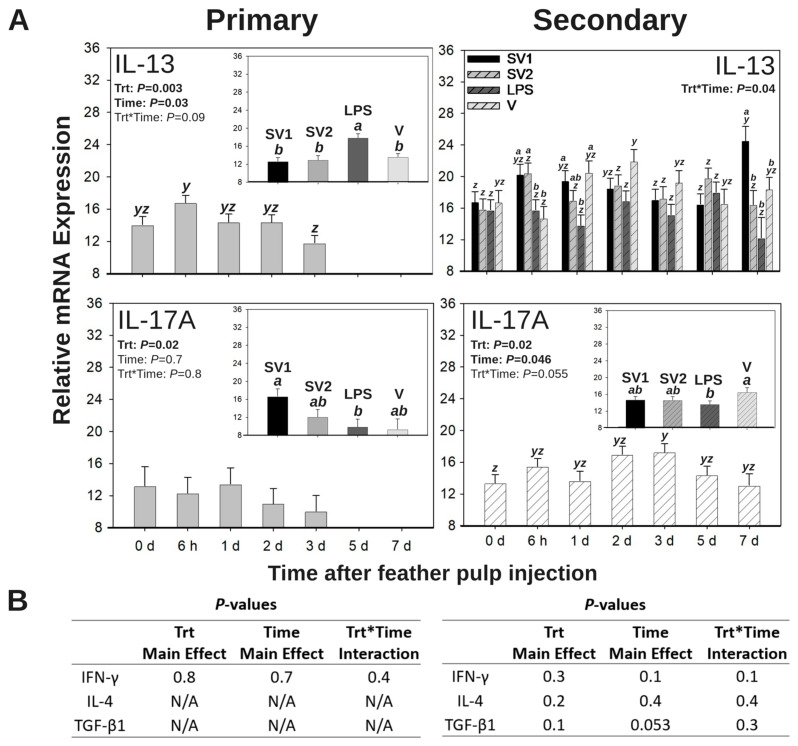
Effector T cell cytokine mRNA expression profiles in growing feather (GF) pulps following primary or secondary intradermal (i.d.) injection of commercial *Salmonella* bacterin vaccines or vaccine components in Light-Brown Leghorn pullets. For the primary i.d. administration, 14-week-old pullets were i.d. injected into the GF-pulp with *Salmonella* vaccine treatments (trt): SV1, SV2, SE lipopolysaccharide (LPS), or water–oil–water emulsion vaccine vehicle (V) (10 μL of injected/GF, 12 GF/pullet, 120 μL of total injection volume). For the secondary i.d. administration, the same trts were subcutaneously (s.c.) administered (0.5 mL) in the nape of the neck of 14-week-old pullets followed by i.d. GF-pulp injection with their respective s.c. trt at 19 weeks of age (21 GF/pullet, 10 μL/GF, 210 μL total injection volume). Injected GF from primary and secondary i.d. vaccinations were collected before (0 d) and at 6 h, 1 d, 2 d, 3 d, 5 d, and 7 d post-injection (p.i.). Cytokine mRNA expression was measured by quantitative RT-qPCR. Data were analyzed by two-way ANOVA (trt, time, and their interactions), and presented as mean 40 − ∆C_T_ ± SEM for (**A**) IL-13 and IL-17A, and (**B**) IFN-γ, IL-4, and TGF-β1. *x–z*: Timepoints that do not share the same superscript are statistically different; *a*,*b*: Trts that do not share the same superscript are statistically different; Trt*time interactions are denoted above each bar in the figure where timepoints that do not share the same superscript (*x–z*) are statistically different, and treatment Groups that do not share the same superscript (*a*,*b*) are statistically different.

**Table 1 vaccines-13-00311-t001:** Experimental designs for the evaluation of local cellular (growing feather pulp) and systemic humoral (blood) primary and secondary immune responses to autogenous *Salmonella* bacterin vaccines or vaccine components.

	Primary Vaccination			Secondary Vaccination		
Group 1: ^1^	I.d. GF-pulp injection ^3^					
	Trt ^5^	Age	Injection Vol. per Bird			
	SV1	14 wk	120 μL			
	SV2					
	LPS					
	V					
Group 2: ^2^	S.c. vaccination ^4^			I.d. GF-pulp injection		
	Trt	Age	Injection Vol. per Bird	Trt	Age	Injection Vol. per Bird
	SV1	14 wk	500 μL	SV1	19 wk	210 μL
	SV2			SV2		
	LPS			LPS		
	V			V		

^1^ Group 1: 14-week-old Light-Brown Leghorn (LBL) pullets underwent a primary intradermal (i.d.) injection with *Salmonella* vaccines (SV1 or SV2) or vaccine components (lipopolysaccharide/LPS, or vaccine vehicle/V) via growing feather (GF) pulp injection. ^2^ Group 2: 14-week-old LBL pullets underwent a primary subcutaneous (s.c.) vaccination with *Salmonella* vaccines (SV1 or SV2) or vaccine components (LPS or V) followed by a secondary i.d. GF-pulp injection with the respective primary vaccination treatment (trt) at 19 weeks old. ^3^ Intradermal (i.d.) GF-pulp injection with 10 μL of a vaccine treatment (10 μL injection/GF; Group 1: 12 GF/bird, Group 2: 21 GF/bird). ^4^ Subcutaneous (s.c.) vaccination in the nape of the neck with 500 μL of a vaccine treatment. ^5^ Vaccine trt were autogenous *S*. Enteritidis (SE)-based commercial *Salmonella* bacterins: SV1 = *Salmonella* Vaccine 1 (10^8^ cells/dose; 180,000 endotoxin units/mL; *n* = 4 bird/trt/group), SV2 = *Salmonella* Vaccine 2 (10^8^ cells/dose; 68,000 endotoxin units/mL; *n* = 4 bird/trt/group), LPS = SE lipopolysaccharide (180,000 endotoxin units/mL; *n* = 4 bird/trt/group), and V = water–oil–water emulsion vehicle (*n* = 3 bird/trt/group) used to suspend SV1, SV2, and LPS vaccine treatments.

**Table 2 vaccines-13-00311-t002:** Immunofluorescent-conjugated antibody combinations for direct-immunofluorescent staining of growing feather pulp cell suspensions for cell population analysis by flow cytometry.

Stain Combinations	Antibody ^1^	Label ^2^	Detected Leukocyte
A	mac-MHCII	FITC	Antigen-presenting cells
	mac-KUL01	PE	Monocytes/Macrophages
	mac-CD45	SPRD	All leukocytes
B	mac-CD45	FITC	All leukocytes
	mac-Bu-1	PE	B lymphocytes
	mac-CD3	SPRD	All T lymphocytes
C	mac-TCRgd	FITC	γδ T lymphocytes
	mac-CD4	PE	CD4^+^ T lymphocytes
	mac-CD8	SPRD	CD8α^+^ T lymphocytes

^1^ mac = mouse-anti-chicken monoclonal antibodies (Southern Biotech, Birmingham, AL, USA). ^2^ FITC = fluorescein isothiocyanate; PE = phycoerythrin; SPRD = SpectralRed^TM^ conjugate. Heterophils (avian neutrophils) analyzed based on forward- and side-scatter distributions (size and granularity characteristics) of the CD45^+^ cells [33].

## Data Availability

The data presented in this study are available in this article and Appendix A.

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
