# Peer review of "Inflammatory and Humoral Immune Responses to Commercial Autogenous Salmonella Bacterin Vaccines in Light-Brown Leghorn Pullets: Primary and Secondary Vaccine Responses"

_vaccines, 2025, doi:10.3390/vaccines13030311_

Round 1
Reviewer 1 Report
Comments and Suggestions for Authors
The manuscript describes changes in inflammatory cytokines and humoral immunity in response to autogenous Salmonella bacterin vaccines (SV1 or SV2) in a prime-boost study, focusing on injection site inflammation caused by the water-oil-water adjuvant formulation. Some issues in the manuscript need to be addressed before publication
Major
The main concern with this manuscript lies in its substantial overlap with the authors' previous work (Santamaria et al. [15]). While the stated objective is to address unanswered questions from that study, much of the experimental design and findings are similar, raising the question of whether the results would have been more appropriately published as a single comprehensive paper rather than two separate manuscripts. A clearer and more detailed explanation of how this work builds upon and extends Santamaria et al. [15] would strengthen its contribution.
-In Introduction, rather than focusing solely on answering questions from prior work, the manuscript would benefit from a broader discussion of other relevant studies in the field.
-the lack of significant changes in key cytokines such as IL-12α and TNF-α raises concerns about whether the vaccines effectively stimulate a robust Th1 immune response. Interpretation of the results is further complicated by the water-oil-water emulsion (V) vehicle, which itself triggers substantial immune responses (e.g., CD4+ and γδ T cell recruitment) independent of the Salmonella antigen.
-The small sample sizes (3-4 pullets per group) is another issue that may reduce statistical power of this study. The study is also limited to a 28-day observation period, restricting insights into long-term immune memory and sustained vaccine efficacy.
- There are inconsistencies in temporal immune responses, such as macrophage levels dropping below baseline following secondary vaccination, which require further clarification.
Minor
-Title can be revised e.g.
“Inflammation and Humoral Immunity to Autogenous Salmonella Vaccines in Light-brown Leghorn Pullets” or “Local Inflammation and Humoral Immunity Induced by Autogenous Salmonella Bacterin Vaccines in Leghorn Pullets: Primary and Secondary Vaccination Responses”
-The abstract needs revision to improve clarity and organization, ensuring that the main findings are clearly highlighted. The analysis methods are not essential for this section and can be omitted.
-Lines 20-24 revise as follow “Elevated heterophil and macrophage levels, along with IL-6 and IL-8 upregulation, peaked at 6 h post-primary injection and remained elevated for up to 3 days after secondary administration, with prolonged IL-17A expression.
-Lines 30-32 revise as follow “Further research is needed to determine whether prolonged inflammation influences adaptive immune responses in eliminating Salmonella infection”
-Introduction could benefit from clearer structure and some revisions to improve clarity, readability, and scientific precision.
-Table 3. Can be moved to supplementary file.
- The provided figures need improvements in design, including adjustments to font and color.
- The number of samples (n) should be clearly indicated to better present the results.
-The main concern mentioned above can be addressed in the Discussion section
Author Response
Dear Vaccines Editors and Manuscript Reviewers,
The co-authors of the submitted manuscript “Local Cellular and Systemic Humoral Immune Responses to Autogenous Salmonella Bacterin Vaccines Following a Primary or a Secondary Vaccination in Light-brown Leghorn Pullets” express gratitude for and recognition of all revisions recommended by all reviewers.
I will proceed point-by-point to address the comments, beginning with “Major Issues” first:
- “The main concern with this manuscript lies in its substantial overlap with the authors' previous work (Santamaria et al. [15]).”
Thank you for the comment; the authors have attempted to address concerns with potential overlaps with Santamaria et al. (2024). Specifically, a note was added in the introduction that states that the goal of this paper was to compare the impact of high- and low-doses of a commercially available Salmonella vaccine on local leukocyte levels after vaccine administration (Line 85). The authors kept the summary of Santamaria’s study in the introduction since the results of this paper build on the findings of previous work. A new section was provided (line 96) to delineate the gaps in knowledge that existed after the completion of the work by Santamaria, and in this manuscript, we have reported our assessments and findings attempting to bridge that gap.
- “In Introduction, rather than focusing solely on answering questions from prior work, the manuscript would benefit from a broader discussion of other relevant studies in the field.”
Thank you for the suggestion; several “sections” of text have been added to the introduction to provide a broader discussion of relevant topics regarding Salmonella bacterin research in poultry. Specifically, sections discussing details about the efficacy of live- and killed-Salmonella vaccination programs (lines 44-51), the benefits of a final bacterin vaccination (lines 52-62), and concerns with severe injection site reactions to bacterin vaccines in oil adjuvants in poultry flocks (lines 62-71). With the addition of this information in the introductions, the authors believe that it provides a decent segway into the study by Santamaria and addresses why further studies needed to be conducted following that research study.
- “The lack of significant changes in key cytokines such as IL-12α and TNF-α raises concerns about whether the vaccines effectively stimulate a robust Th1 immune response.”
Thank you for the comment; the authors were surprised by this finding as well. We expected to observe a robust Th1 response since Salmonella is an intracellular pathogen, but that simply was not the case for this study. The cell-mediated immune response that was observed polarized to a Th17-type response at the site of injection along with indices of chronic inflammation. This was noted in the abstract (line 29) and discussed in sections 4.3 and 4.4 of the discussion (especially lines 805-812 and 831-844). We also speculate about why a Th1 response was not observed from lines 911-915. It is possible that the lack of a Th1-response in this study is due to the Salmonella vaccine being a killed bacterin (rather than live attenuated).
- “Interpretation of the results is further complicated by the water-oil-water emulsion (V) vehicle, which itself triggers substantial immune responses (e.g., CD4+ and γδ T cell recruitment) independent of the Salmonella antigen”
Thank you for the comment; the authors were also surprised by this finding. It is apparent that the Salmonella bacterin has some form of a masking effect on the actions produced by the vaccine vehicle at a local level. However, there was clear evidence of an adaptive, cell-mediated response due to the evidence of a T cell-dependent B cell activation as seen by the IgM-to-IgG antibody shift. To further address these concerns, We have added a paragraph in discussion section 4.4 that specifically discusses B cell activation that likely occurred in secondary lymphoid organs after vaccination (lines 878-897).
- The small sample sizes (3-4 pullets per group) is another issue that may reduce statistical power of this study. The study is also limited to a 28-day observation period, restricting insights into long-term immune memory and sustained vaccine efficacy.
Thank you for the comment and concern regarding sample size. To address this concern, the authors conducted a post hoc power analysis for sample size (target power: 80%, Cohen’s d effect size = 0.3). When assessing the minimum treatment replicate value for Salmonella-specific antibody levels in this study (4 groups X 3 pullets per group X 8 timepoints), the sample power analysis is 83.2%. When assessing the minimum treatment replicate value for local leukocyte- and cytokine mRNA expression-levels in this study (4 groups X 3 pullets per group X 8 timepoints, 120 samples total), the sample power analysis is 80.0%. When assessing the maximum treatment replicate value for local leukocyte- and cytokine mRNA expression-levels in this study (4 groups X 4 birds per group X 7 timepoints), the sample power analysis was 83.3%.
Previous studies that used the growing feather pulp rejection model also required power analysis to determine sample size. These analyses determined that the ideal treatment replicate was 4-8 birds per treatment per sample collection timepoint. Determining the sample number depends on the number of sample collection timepoints to be conducted and the number of treatments to be applied. We have also found that the growing feather pulp injection model is highly replicable in most studies, which makes this methodology an excellent model when considering animal welfare.
- There are inconsistencies in temporal immune responses, such as macrophage levels dropping below baseline following secondary vaccination, which require further clarification.
Thank you for the comment! Since these local leukocyte levels are presented as a percentage of total growing feather pulp cells, it is entirely possible for macrophage levels to fall below baseline following the secondary vaccination. The feather pulp is a live and complex tissue. It is possible that macrophages significantly drop at the site of injection because they are traveling to secondary lymphoid organs in participating in antigen presentation to lymphocytes (line added 733-735).
Next, I will discuss “Minor Issues” addressed by the reviewer:
- Title can be revised e.g. “Inflammation and Humoral Immunity to Autogenous Salmonella Vaccines in Light-brown Leghorn Pullets” or “Local Inflammation and Humoral Immunity Induced by Autogenous Salmonella Bacterin Vaccines in Leghorn Pullets: Primary and Secondary Vaccination Responses”
Thank you for the recommendation. The title has been revised to “Inflammatory and Humoral Immune Responses to Commercial Autogenous Salmonella Bacterin Vaccines in Light-brown Leghorn Pullets: Primary and Secondary Vaccine Responses.”
- The abstract needs revision to improve clarity and organization, ensuring that the main findings are clearly highlighted. The analysis methods are not essential for this section and can be omitted.
Thank you for the recommendation. The abstract has been revised (278 words), and the analysis methods were removed. Abstract titles (Background/Objectives, Methods, Results, Conclusions) were also added into the abstract text.
- Lines 20-24 revise as follow “Elevated heterophil and macrophage levels, along with IL-6 and IL-8 upregulation, peaked at 6 h post-primary injection and remained elevated for up to 3 days after secondary administration, with prolonged IL-17A expression.
Thank you for the recommendation. We have assessed the change and compromised with the following statement, “Primary vaccine administration increased local heterophil- and macrophage-levels and increased IL-6 and IL-8 mRNA expressions at 6h p.i., independent of treatment. Secondary administration extended these local immune activities through 3d p.i. and included prolonged IL-17A mRNA expression.”
- Lines 30-32 revise as follow “Further research is needed to determine whether prolonged inflammation influences adaptive immune responses in eliminating Salmonella infection”
Thank you for the recommendation. The concluding statement now reads “Further research is needed to determine if extended inflammation influences adaptive immune responses in eliminating Salmonella infection.”
- Introduction could benefit from clearer structure and some revisions to improve clarity, readability, and scientific precision.
Thank you for the comment. The introduction has been edited accordingly (please see responses to previous introduction suggestions).
- Table 3. Can be moved to supplementary file.
Thank you for the suggestion. The table has been modified to a supplementary file.
- The provided figures need improvements in design, including adjustments to font and color.
Thank you for the comment. We have made edits accordingly, but we may not have addressed all issues as envisioned.
- The number of samples (n) should be clearly indicated to better present the results.
Thank you for the suggestion. “n” has been added every figure or figure caption.
- The main concern mentioned above can be addressed in the Discussion section
Thank you. Please see previous notes to address concerns with the discussion section.
Again, myself and my fellow co-authors thank you for your time and dedication to reviewing our manuscript. We look forward to your feedback on our revisions.
Thank you for your time.
Regards,
Chrysta Beck

Reviewer 2 Report
Comments and Suggestions for Authors
Dear authors
I hope this finds you all well. Regarding the review of manuscript number vaccines-3509836, entitled "Local Cellular and Systemic Humoral Immune Responses to Autogenous Salmonella Bacterin Vaccines Following a Primary or a Secondary Vaccination in Light-brown Leghorn Pullets". It is indeed interesting research, but some important comments should be replied
Comments
- What were the serotypes of different Salmonella vaccines used?
- The quantification of ELISA titers of specific Salmonella Immunoglobulins in Blood Plasma was based on S. Enteritidis. Did this match with the serotype of Salmonella vaccines used???
- More details are needed about the 4 Salmonella vaccine formulations used (SV1=Salmonella Vaccine 1 (180,000 endotoxin units/mL), SV2=Salmonella Vaccine 2 (68,000 endotoxin units/mL), LPS=SE lipopolysaccharide (180,000 endotoxin units/mL), and V=water-oil-water emulsion vaccine). Did you use a commercial or homemade water-oil-water emulsion vaccine??
- The total number in both groups and subgroups are not clear. Please explain this in M & M.
Author Response
Dear Vaccines Editors and Manuscript Reviewers,
The co-authors of the submitted manuscript “Local Cellular and Systemic Humoral Immune Responses to Autogenous Salmonella Bacterin Vaccines Following a Primary or a Secondary Vaccination in Light-brown Leghorn Pullets” express gratitude for and recognition of all revisions recommended by all reviewers.
I will proceed point-by-point to address the comments:
- “What were the serotypes of different Salmonella vaccines used?”
Thank you for the comment. The formulations of the vaccines were proprietary, and the identity of the test vaccines are not currently disclosed. The only information that we have is about the endotoxin unit concentration. A note was added to further clarify this information (line 135-138).
- “The quantification of ELISA titers of specific Salmonella Immunoglobulins in Blood Plasma was based on S. Enteritidis. Did this match with the serotype of Salmonella vaccines used???”
Thank you for the comment. A note was added to further clarify this information, but the formulations were proprietary. Unfortunately, we do not know the details of all Salmonella serovars included in the vaccines, but we do know that both SV1 and SV2 contained S. Enteritidis (line 135-138).
- “More details are needed about the 4 Salmonella vaccine formulations used (SV1=Salmonella Vaccine 1 (180,000 endotoxin units/mL), SV2=Salmonella Vaccine 2 (68,000 endotoxin units/mL), LPS=SE lipopolysaccharide (180,000 endotoxin units/mL), and V=water-oil-water emulsion vaccine). Did you use a commercial or homemade water-oil-water emulsion vaccine??”
Thank you for the comment. A note was added to further clarify this information, but the formulations were proprietary (line 135-138). The emulsion was commercial, and the vaccines were already emulsified and ready-to-use when we received them.
- “The total number in both groups and subgroups are not clear. Please explain this in M & M”
Thank you for the comment. We have clarified the total number of birds per treatment in the materials and methods (highlighted throughout the document).
Again, myself and my fellow co-authors thank you for your time and dedication to reviewing our manuscript. We look forward to your feedback on our revisions.
Thank you for your time.
Regards,
Chrysta Beck

Reviewer 3 Report
Comments and Suggestions for Authors
This manuscript continues the previous manuscript, by the same author, reporting the immune response, local and systemic, after vaccination pullets with autogenous Salmonella enteritidis bacterin by intradermal injection into the growing feather pulp. The authors followed the primary response after vaccination at 14 weeks old and a secondary immunization at 19 weeks. Different groups were injected with either 180,000 endotoxin units/mL, or 68,000 endotoxin units/mL, and their response was compared with a group of pullets injected with LPS and a group injected with water in oil in water group. The results are well-presented and the discussion is clear. The main objective was to show that this method of vaccination causes no local damage and provokes a good local and systemic immune response.
Author Response
Dear Vaccines Editors and Manuscript Reviewers,
The co-authors of the submitted manuscript “Local Cellular and Systemic Humoral Immune Responses to Autogenous Salmonella Bacterin Vaccines Following a Primary or a Secondary Vaccination in Light-brown Leghorn Pullets” express gratitude for and recognition of all revisions recommended by all reviewers.
Thank you for your comments and feedback on our submitted manuscript. Myself and my fellow co-authors thank you for your time and dedication to reviewing our manuscript, and we look forward to any further feedback you may have regarding our revisions.
Thank you for your time.
Regards,
Chrysta Beck

Round 2
Reviewer 1 Report
Comments and Suggestions for Authors
Accept.